# Quantification of Temperature Dependence of Hydrogen Embrittlement in Pipeline Steel

**DOI:** 10.3390/ma12040585

**Published:** 2019-02-15

**Authors:** Xiao Xing, Jiayu Zhou, Shouxin Zhang, Hao Zhang, Zili Li, Zhenjun Li

**Affiliations:** 1College of Pipeline and Civil Engineering, China University of Petroleum (East China), Qingdao 266500, China; S17060737@s.upc.edu.cn (J.Z.); b15060157@s.upc.edu.cn (S.Z.); cygcx@163.com (Z.L.); 2Department of Chemical and Materials Engineering, University of Alberta, Edmonton, AB T6G 1H9, Canada; Hao7@ualberta.ca; 3Petrochina West Pipeline Company, Wulumuqi 830001, China; xbgdlizhj@petrochina.com.cn

**Keywords:** hydrogen embrittlement, hydrogen diffusion, stress rupture, fatigue

## Abstract

The effects of temperature on bulk hydrogen concentration and diffusion have been tested with the Devanathan–-Stachurski method. Thus, a model based on hydrogen potential, diffusivity, loading frequency, and hydrostatic stress distribution around crack tips was applied in order to quantify the temperature’s effect. The theoretical model was verified experimentally and confirmed a temperature threshold of 320 K to maximize the crack growth. The model suggests a nanoscale embrittlement mechanism, which is generated by hydrogen atom delivery to the crack tip under fatigue loading, and rationalized the Δ*K* dependence of traditional models. Hence, this work could be applied to optimize operations that will prolong the life of the pipeline.

## 1. Introduction

Hydrogen embrittlement [1,2] is the most severe degradation mechanism in buried pipeline steel and is an issue involving the loading, environment, and physical property of the steel [3,4,5,6]. Temperature is a determining factor in hydrogen diffusivity and bulk hydrogen concentration *c_o_* and, thus, also determines the level of hydrogen embrittlement [7,8,9]. Hence, crack propagation in pipeline steel has a strong temperature dependence. However, traditional models (like Paris’ law) only quantify the relationship of the crack growth with the stress intensity range (Δ*K*) [10], or verify the correlation of crack growth with loading frequency *f* [11,12,13,14]. Additionally, no model is available that quantifies the effect of temperature on crack growth.

When hydrogen is not present in steel, the toughness of the steel would increase with temperature, meanwhile, the hardness would decrease. Specifically, the crack resistance of steel would increase with temperature [15,16]. However, when hydrogen is introduced, the temperature dependence of the crack resistance of steel remains debatable, and further studies are required in this field. Therefore, the hydrogen diffusion model has been applied to quantify the effect of temperature. Hydrogen atoms are driven by hydrostatic stress to accumulate around the crack tip [17,18]. The trapped hydrogen atoms near a crack lower the free surface energy [19], thus enhancing cleavage-like failure. This postulation is the hydrogen-enhanced decohesion (HEDE) mechanism [20,21,22]. The hydrogen-enhanced local plasticity mechanism (HELP) [23,24,25,26] can be interpreted in a way that hydrogen atoms enhance the dislocation generation and emission, thus increasing the local plasticity and leading to the subsequent failure by exhaustion of the material strain capability. These two mechanisms provide a cleavage mechanistic framework for predictive models: (1) hydrogen atoms will be trapped in plastic deformations and will diffuse into the plastic zone and saturate plastic defects [27,28,29]; (2) the hydrogen atoms will aggregate at the crack tip after saturating the plastic zone, diminishing the energy for cleavage [18]; and (3) as the atomic hydrogen concentration (an atomic ratio of number of hydrogen atoms to iron atoms) at the crack tip approximates 1, every bond between adjacent iron atoms would be weakened by the hydrogen atom, and the bindings are easily broken to form free surfaces fully covered by hydrogen atoms [17]. Typically, the temperature will not affect the plastic zone size because yield strength has little temperature dependence when compared with hydrogen diffusivity and concentration [11]. As the temperature rises from 293 to 373 K, the yield strength only changes one percent [30]. Therefore, the temperature dependence of hydrogen diffusivity and equilibrium are critical factors that should be determined to quantify the temperature dependence of hydrogen embrittlement. 

A crack growth model based on hydrogen diffusion [17] can be described as
(1)$$\dot{a}=\frac{d\Delta a}{dt}=\frac{4}{\pi }\frac{D\Omega_{H}}{k_{B}T}\frac{\left(1+v\right)K_{I}}{3\sqrt{2\pi }}\left(\frac{\pi c_{o}}{a_{o}}\right)\Delta a^{-1/4},$$
where $\dot{a}$ is the crack growth rate; Δ*a* is the crack length change; *D* is the hydrogen atom diffusivity; m^2^/s; Ω*_H_* is the partial volume of hydrogen atom; *v* is Poisson’s ratio; *c_o_* is the bulk equilibrium hydrogen atomic concentration; *a_o_* is the original length of the crack; *k_B_* is the Boltzmann constant; and *T* is temperature. This model establishes a quantitative relationship between the temperature and crack growth rate because both *D* and *c_o_* are dependent on temperature *T*. However, this model is only applicable in static loading and in a brittle condition because of the plastic zone [31,32] which is generated in cyclic loading, as the aggregation of hydrogen atoms by plastic deformations is ignored in this model. However, this model involves the physical properties of the materials and environmental parameters, as well as the loading factors, whereas only the loading factors are considered in traditional models. To improve the theoretical model so that it can be applied in industrial operations, hydrogen-diffusion related crack growth theory was applied to a new model where hydrogen atoms saturate the plastic zone before they diffuse to the crack tip. As Figure 1 shows, the stress and dislocation density are small outside the plastic zone when compared with inside, and the hydrogen concentration outside the plastic zone is approximated to be the bulk equilibrium concentration *c_o_*. An illustrated annulus region outside the plastic zone will offer hydrogen atoms during loading and deplete partial hydrogen atoms during unloading. If the annulus region is large enough (the resource of hydrogen atoms is enough) and the loading time is long enough, the crack growth rate is maximized as hydrogen atoms have enough time to saturate the plastic zone during loading. As the hydrogen concentration approximates unity at the crack tip, the free surfaces are generated.

## 2. Temperature Dependence of Critical Loading Frequency

Critical loading frequency was identified back in the 1880s and is a frequency in the loading spectra below which the crack growth rate reaches the maximum and remains constant [33]. This threshold is only established as hydrogen atoms exist in the specimen. It is natural to associate this threshold with hydrogen diffusion and aggregation. The movement of hydrogen atoms dependent on axial stress *F_r_*, and *F_r_* near the crack tip, is related to the hydrostatic stress $\sigma^{hyd}$, $F_{r}=-\Omega_{H}\nabla_{r}\sigma^{hyd}$. $\Omega_{H}$ is the partial volume of the hydrogen atom and the average velocity of the hydrogen atom is defined as [28]
(2)$${\bar{V}}_{r}=\frac{\int_{r_{p}}^{r_{p}+r_{eq}}\int_{K_{\min}}^{K_{\max}}\frac{DF_{r}}{k_{B}T}dK_{I}dr}{\left(K_{\max}-K_{\min}\right)r_{eq}},$$
where *r_p_* is the size of the plastic zone and *r_eq_* is the size of an annulus region around the plastic zone to supply sufficient hydrogen atoms into the plastic zone. The saturation number of hydrogen atoms *N*(*K_I_*) is related to the stress, the size of plastic zone, and the bulk equilibrium hydrogen concentration, *c_o_*. It is also equal to the number of hydrogen atoms that the virtual hydrogen supply area, encircled by the dotted line in Figure 1, can offer:(3)$$N\left(K_{\max}\right)-N\left(K_{\min}\right)=\frac{2\pi c_{o}l_{z}}{a_{o}^{3}}\left({\left(r_{p}+r_{eq}\right)}^{2}-r_{p}^{2}\right),$$
where
(4)$$N\left(K_{I}\right)=\int_{0}^{r_{p}}\left(c_{o}\exp\left(\frac{\sigma^{hyd}\Omega_{H}}{k_{B}T}\right)2\pi rl_{z}/\left(a_{o}^{3}/2\right)\right)dr,$$
where *l_z_* is the thickness of specimen, and the time that requires hydrogen atoms to saturate the plastic zone as the stress intensity rises from *K*_min_ to *K*_max_ is $t_{c}=\frac{r_{eq}}{{\bar{V}}_{r}}$. The threshold for the loading frequency is *f_c_* = 1/(2*t_c_*). This threshold is sensitive to temperature because some of the parameters have a strong temperature dependence. For example, the diffusivity of hydrogen atoms is in an exponential function with T and is expressed as *D* = *D_o_* × 10^−8^ × EXP(−EkBT). The diffusion coefficients in iron have a large scatter because hydrogen atoms can be trapped by impurities. Typical *D_o_* values can range from 3.35 × 10^−8^ to 2.2 × 10^−7^ m^2^/s, and E is the diffusion barrier which ranges from 0.035 to 0.142 eV. The specimen was assumed to have a bcc structure, hence, the theoretical *D* expression, which is deduced from first principles in the bcc lattice, is shown as [34]
(5)$$D=1.5\times 10^{-7}\exp\left(-0.088\;eV/k_{B}T\right)\;\;\;\;m^{2}s^{-1}.$$

The predicted threshold of loading frequency (*f*_critical_) corresponds to a scenario where the plastic zone is saturated with hydrogen atoms, and it is also expected that less time is required to saturate the plastic zone if the movement of hydrogen atoms (${\bar{V}}_{r}$) is enhanced. The inset of Figure 2 shows that the predicted velocity of the hydrogen atoms increases with temperature, and the prediction was verified in Figure 4 in the next section. Basically, more hydrogen atoms can aggregate in the plastic zone during the same time interval and under the same stress intensity as *T* is increased. In other words, the time required to saturate the plastic zone is smaller and the temperature is higher under the same loading condition. In Figure 2, the filled symbols were tested for critical loading frequency in HY-130 steel and the open symbols are the predicted values. The results showed rough agreement and verified the temperature enhancement to critical loading frequency. The predicted values and empirical values did not converge as the stress intensity was large (Δ*K* = 30.8 MPa * m^0.5^), because the space between the adjacent iron atoms was enlarged as the stress intensity increased. Thus, hydrogen diffusion was enhanced because of the extra volume induced by a larger stress intensity [35]. This kind of enhancement is negligible in this calculation, however, predictions under common stress intensities (Δ*K* = 14 or 20.9 MPa * m^0.5^) showed good agreement with the empirical values.

## 3. Temperature Dependence of Hydrogen Concentration

The hydrogen-assisted failure mechanism can be facilitated either by increasing the hydrogen concentration or increasing the hydrogen movement. The temperature dependence of hydrogen concentration and diffusion can be tested with Devanathan–Stachurski double electrolysis cells. An X80 steel specimen (the hydrogen density and diffusivity are not related to the strength of the steel, so the X80 test results can be applied to other steels) with a thickness of 1200 µm and diameter of 35 mm is shown in Figure 3a. The sample was sealed by polytetrafluoroethylene with 740 silica gel on one surface and was electroplated with nickel on the other surface to prevent the recombination of the hydrogen atoms. After the electroplating was completed, the polytetrafluoroethylene was removed from the specimen and the specimen was cleaned with deionized water.

Afterwards, the dried steel specimen was located between two electrolysis cells as shown in Figure 3b. The corrosion test on the naked side was performed in a near neutral pH solution which was employed to charge the hydrogen atoms under a polarized electrical potential of 300 mV. Initially, the hydrogen atoms were generated on the naked side and moved to the other side of the specimen afterwards. A PARSTAT2273 was applied to test the current density of the specimen on the test side, which was covered with nickel. The tested current density, as shown in Figure 4, is related to the hydrogen concentration. As the current approaches a constant value, the surface hydrogen concentration approximates the bulk equilibrium hydrogen concentration.

The facilitating effect of temperature on hydrogen movement was confirmed by the results shown in Figure 4. The sharp rise of the current density could be detected and recorded in a shorter time at high temperature. To be specific, the length of the initial horizontal line was negatively proportional to the temperature, which indicates that less time was spent by the first hydrogen atom travelling to the testing side. The positive correlation of hydrogen concentration with temperature was verified using Figure 4. As osmotic current densities of hydrogen atoms were tested at different temperatures, there was a terrace value for each temperature where the current density stayed constant, and this current density corresponded to the bulk hydrogen concentration in the steel. The relationship of current density $i_{\infty }$ with bulk hydrogen concentration is
(6)$$c_{o}=\frac{i_{\infty }l_{z}}{FD},$$
where *F* is the Faraday constant and *D* is the diffusivity of the hydrogen atoms. The current density $i_{\infty }$ increased remarkably with temperature, hence *c_o_* increased with temperature and a specific *c_o_* value could be calculated with Equation (6) at different temperatures. However, the equilibrium hydrogen concentration near a crack has an exponential relation with *T*, as shown in Equation (7), which suggests that increasing temperature may diminish the hydrogen concentration near a crack.

(7)$$c_{eq}=c_{o}\exp\left(\frac{4\left(1+v\right)K_{I}\Omega_{H}}{3\pi k_{B}T\sqrt{2\pi r}}\right)$$

Consequently, the experiments suggest that increasing temperature will facilitate the movement of hydrogen atoms and bulk hydrogen concentration, however, Equation (7) suggests that as temperature increases the *c_o_*, it still diminishes the exp(1/*T*) set. Hence, the specific value of *c_eq_* should be calculated as determining the crack growth rate.

## 4. Temperature Dependence of Crack Growth Rate

To quantify the temperature dependence of crack growth, a model based on hydrogen-enhanced decohesion (HEDE) was developed. Here, we employed the criteria that when the atomic ratio of a hydrogen atom ahead of a crack reached 1, the free surfaces formed, and the crack propagated. The corresponding free surface length at different stress intensities can be shown as
(8)$$L\left(K_{I}\right)={\left[\frac{4\left(1+v\right)\Omega_{H}}{3\pi k_{B}T\sqrt{2\pi }\ln\left(1/c_{o}\right)}\right]}^{2}K_{I}^{2},$$
and the HEDE model can be indicated as $\left[L\left(K_{\max}\right)-L(K_{\min})\right]/{\left(f/f_{critical}\right)}^{0.1}$
(9)$${\left(\frac{da}{dN}\right)}_{HEDE}=A\left(\frac{1+R}{1-R}\right)\frac{\Delta K^{2}}{{\left(f/f_{critical}\right)}^{0.1}},$$
where *A* is an environmental factor related to temperature and pH, and can be illustrated as $A={\left(\frac{4\left(1+v\right)\Omega_{H}}{3\pi k_{B}T\sqrt{2\pi }\ln\left(1/c_{o}\right)}\right)}^{2}$ [19,27]. The set $\left(\frac{1+R}{1-R}\right)\Delta K^{2}$ is also a component of energy density in the Griffith model where all strain energy is converged to free surface energy [36,37]. The energy density is shown as
(10)$$\Delta s=\frac{1-2v}{4\pi G(T)}\left(\frac{1+R}{1-R}\right)\Delta K^{2}.$$
Hence, this model verified that the Δ*K* dependence of crack growth under the near neutral pH stress corrosion cracking (NNpHSCC) condition is a brittle mechanism and can be applied to rationalize the traditional crack growth models. However, the hydrogen bubble theory [38,39,40], which consumes hydrogen atoms, was neglected in this derivation. Since both mechanisms are related to hydrogen diffusion, the total hydrogen-assisted cracking rate is naturally expected to have a power law relationship with the crack growth rate based on HEDE. The total crack growth rate is ${\left(da/dN\right)}_{T}={\left(da/dN\right)}_{HEDE}^{n}$. As the loading frequency *f* is smaller than *f*_critical_, the plastic zone is saturated with hydrogen atoms and the crack growth rate is not dependent on the loading frequency and reaches the maximum. HY130 steel is a high strength steel and the n value approaches 1. In Figure 5, the experiments were carried out in C-2 (0.0035 KCl, 0.0195 NaHCO_3_, 0.0255 CaCl, 0.0274 MgSO_4_∙7H_2_O, 0.0606 CaCO_3_ g/L) solution. The specimen was gritted to 9 mm thickness and a notch in the compact tension (CT) specimens was orientated in a direction perpendicular to the mode I loading direction. The dimensions of the specimen should be 50 × 50 × 9 mm^3^. The specimen was polished to 600 grit finish and then pre-cracked in air by fatigue to generate a sharp crack from the notch. Afterwards, the specimen was sealed in the test cells and pinhole-loaded with computer load control. The crack growth rate was recorded at an R ratio of 0.25, a loading frequency of 0.1 or 4.9, and the Δ*K* value was constant at 14 MPa * $\sqrt{m}$, and the temperature ranged from 280 to 350 K [31]. The temperature threshold to maximize the crack growth under NNpHSCC is indicated in Figure 5, where *f* is equal to 0.1. Elevating temperature will facilitate hydrogen movement and increase the surface hydrogen concentration, however, the equilibrium hydrogen concentration near a crack has an exponential relation with T. More specifically, elevating temperature will enhance hydrogen generation and diffusion but restrict hydrogen accumulation near defects. Consequently, the threshold was generated in the vicinity of 320 K, as shown by the black symbols in Figure 5. If there is insufficient diffusion time for the hydrogen atoms to accumulate (loading frequency is large), the restriction effects of hydrogen accumulation caused by the increase in temperature are not significant when compared with the enhancements of hydrogen diffusion and generation. Hence, as the loading frequency *f* was bigger than the *f*_critical_, the predicted threshold was not detected, as shown by the red symbols.

## 5. Crack Growth Rate Prediction in Different Steels

In Figure 6, the crack growth rate in X65 was tested at different R ratios and loading frequencies, and the Δ*K* value ranged from 10 to 25 MPa * $\sqrt{m}$ [41]. The yield strength *σ_ys_* of X65 steel was 448 MPa, and was much smaller than that of the HY130 steel. Typically, steel with a smaller yield strength has a larger plastic zone and generates more plastic deformations before failure as the plastic zone size is negatively related to yield strength under the plain strain condition, $r_{p}=\frac{1}{6\pi }{\left(\frac{K_{\max}}{\sigma_{ys}}\right)}^{2}$. The size of the plastic zone is smaller for steel that possesses a higher yield strength. The plastic zone is the area concentrated with defects caused by plastic deformations, which would trap significant amount of hydrogen atoms. The theoretical model was based on HEDE theory; therefore, the trapping effects of plastic deformation-caused defects had been ignored. Hence, for a high strength steel in which the plastic zone is small, the model is more accurate and the fixing parameter n approximates unity. Hence, the n value of X65 steel was smaller than that of HY130. In Figure 6, all predicted crack growth data fit very well with the n value of 0.92.

The same rule can be applied to another steel, X52, which possesses a smaller yield strength when compared with X65. In Figure 7, the R ratio was constant, the loading frequency ranged from 0.001 to 0.005, and the stress intensity was distributed from 15 to 19 MPa * $\sqrt{m}$ [42]. All predictive values were located in the range of error bar of the empirical values and confirmed that all predictions matched well with the experiments with an n value of 0.88. A triangulation rationalized that the brittle crack growth model based on hydrogen embrittlement is applicable in high strength steel. The hypothesis of the model, where a sharp rise of hydrogen concentration causes the formation of a nanoscale hydrogen-rich region in which the hydrogen concentration approached ten thousand times of the bulk equilibrium hydrogen concentration, was verified.

## 6. Conclusions

In the presented study, we tested the temperature dependence of hydrogen generation and diffusion in high strength steel. Based on a mechanism of hydrogen-assisted brittle failure, a theoretical model was developed to predict crack growth rate in different loading spectra and at different temperatures. The model yielded rough agreement with the empirical values in different steels with fixed n values and suggests that the n value in the brittle model is positively correlated with the degree of brittleness of the steel, and is approximately 1 in high strength steel. This conclusion verified the brittle failure mechanism in pipeline steel and suggests that a nanoscale hydrogen-rich region may exist ahead of a crack tip in the NNpHSCC condition.

The model also served to rationalize the empirical models and provide a framework of Δ*K* dependence of crack growth. As the loading frequency was smaller than the *f*_citical_, the temperature threshold to maximize crack growth was predicted in the model and verified by experiment. These results confirmed the predictive capability of the model and suggest that this model can be applied to optimize steel operations.

## Figures and Tables

**Figure 1 materials-12-00585-f001:**
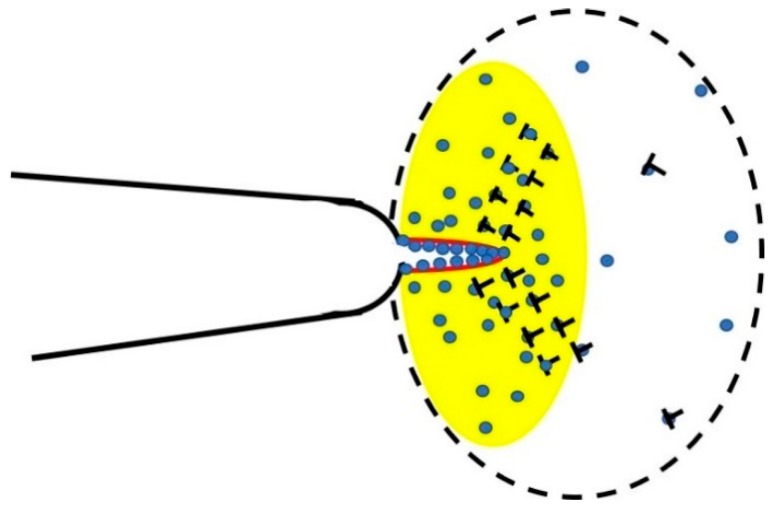
Schematic showing the evolution of a pre-existing microcrack; the yellow circular zone is the plastic zone where the stress is concentrated and plastic deformations (dislocations) are aggregated. The hydrogen atoms need to distribute according to the stress intensity and saturate the dislocations core (T symbols), then diffuse to the crack tip. If the atomic hydrogen concentration ahead of a crack tip approximates 1, free surfaces are generated (red). The stress intensity and dislocation density outside the plastic zone is low, hence, the hydrogen concentration outside the plastic zone approximates the bulk hydrogen concentration, *c_o_*.

**Figure 2 materials-12-00585-f002:**
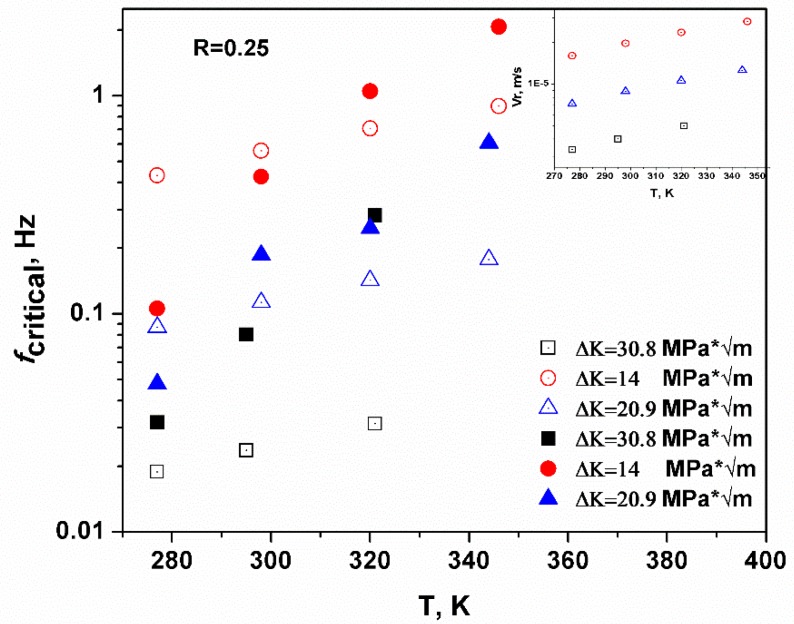
The predicted and tested critical loading frequency versus temperature, where the filled symbols are the tested values in the HY-130 steel, and the open symbols are the predicted values. Both sets of data showed the same tendency, which verified the predictive capability of the model. The inset predicted the hydrogen movement velocity at different temperatures and showed that hydrogen movement was enhanced by temperature [33].

**Figure 3 materials-12-00585-f003:**
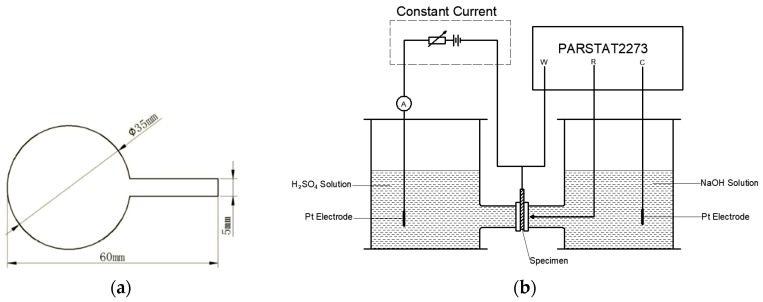
(**a**) The X80 specimen. (**b**) The equipment designed for this study, which consisted of a constant current, the PARSTAT2273, and double electrolysis cells.

**Figure 4 materials-12-00585-f004:**
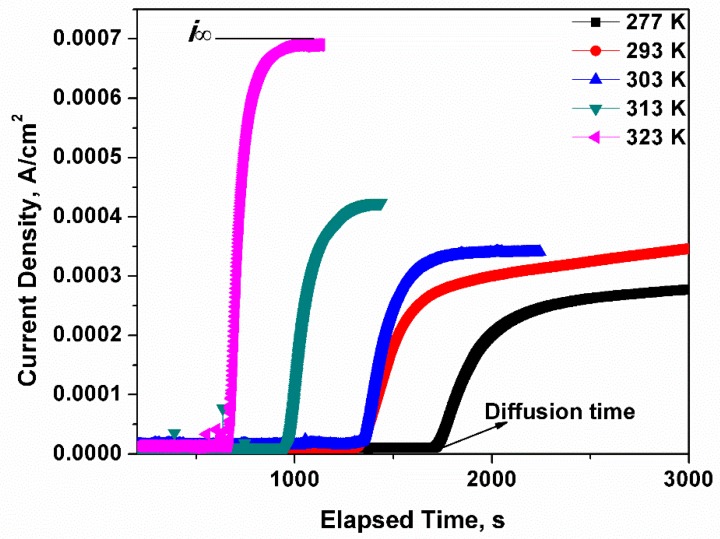
The current density curves of X80 steel. The stable current density increases with the temperature, which means more hydrogen atoms are generated at the surface of the specimen. Namely, the supply of hydrogen atoms increases in the specimen as the temperature increases. As the hydrogen density and diffusivity are not related to the strength of the steel, the X80 test results can be applied to other steels.

**Figure 5 materials-12-00585-f005:**
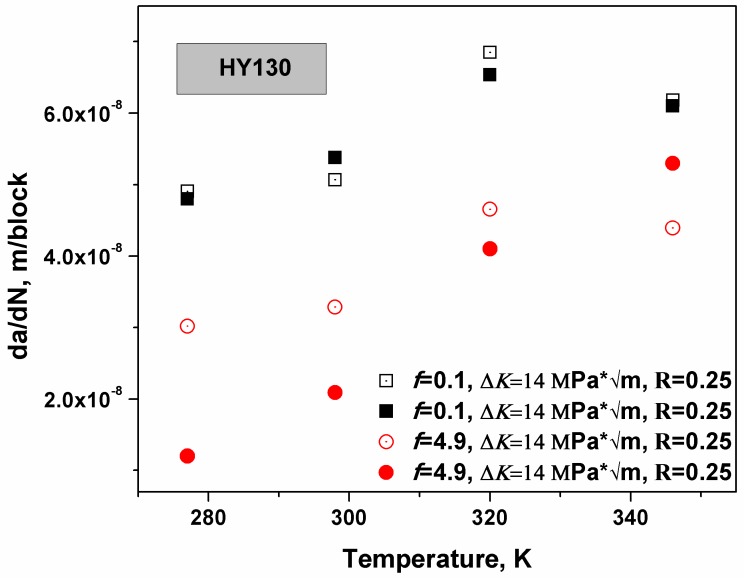
This schematic illustrates the temperature dependence of crack growth in HY130 steel [33]. As *f* < *f*_critical_, the crack growth rate was maximized at 320 K.

**Figure 6 materials-12-00585-f006:**
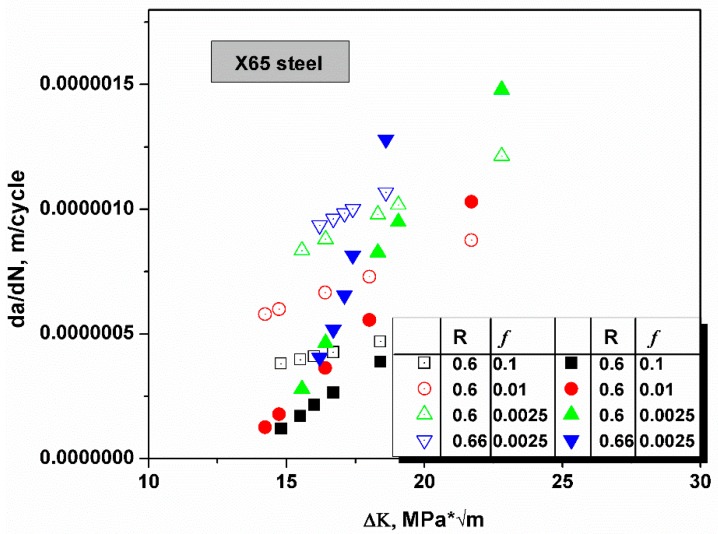
The schematic shows the comparison of the predicted crack growth rates (open symbols) and experimental values [41]. The R ratio and loading frequency *f* are illustrated in the chart. At a smaller R ratio, the predictive accuracy was better as the lattice parameter of the material (this effect is usually negligible) and the diffusivity of H atoms is increased with stress intensity. Predicted values are smaller than the experimental values at a large *K_I_*.

**Figure 7 materials-12-00585-f007:**
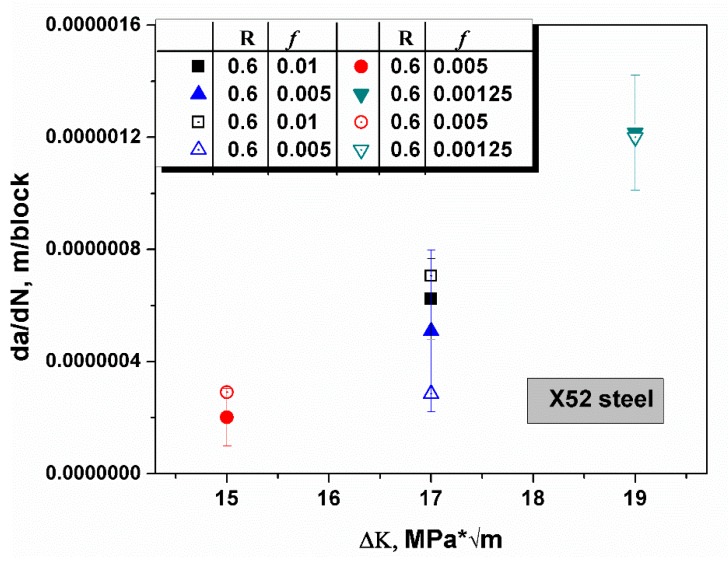
The schematic shows the comparison of the predicted crack growth rates (open symbols) and experimental values in X52 steel [42]. The R ratio and loading frequency *f* are recorded in the chart. The predictive values fit well with the experimental data at an n value of 0.88.

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
