# Peer review of "Quantification of Temperature Dependence of Hydrogen Embrittlement in Pipeline Steel"

_materials, 2019, doi:10.3390/ma12040585_

Round 1
Reviewer 1 Report
The manuscript is organized well. The english part need to be polished before final acceptance.
Author Response
The reviewer wrote: “The manuscript is organized well. The english part need to be polished before final acceptance.”
Thanks for the suggestions. The revised manuscript has been polished and proofread by MDPI English Editing Service.

Reviewer 2 Report
Although the authors did a valuable approach to result in their outstanding modelling work, the quality of presentation is not as competitive as what they did. In other words, it is difficult to catch and understand the key contribution of this manuscript. Therefore, it is highly recommended for the authors to improve the way of their presentation.
The experimental method to measure the temperature dependence of crack growth rate was not very well described. Also, to discuss the role of the temperature depedence of hydrogen-related properties on the crack growth rate, it is necessary to include the temperature dependence data of the crack growth without hydrogen. In that way, the readers would understand the effect of hydrogen and strong need for the modelling.
Also, it is necessary to compare the data from all the different steels and to discuss the origin of difference in them based on the model developed. For example, the difference in crack growth rate under hydrogen absorption between different steels should be explained.
"The criteria of the hydrogen concentration at crack tip approximates one [15], as the free surface forms". Please rephrase the sentence so that the readers can understand the intention of the authors. Also please explain about the unit of 'one'.
NNpHSCC should be defined.
Author Response
The reviewer wrote: “Although the authors did a valuable approach to result in their outstanding modelling work, the quality of presentation is not as competitive as what they did. In other words, it is difficult to catch and understand the key contribution of this manuscript. Therefore, it is highly recommended for the authors to improve the way of their presentation.".”
Thanks for the suggestions. The revised manuscript has been polished and proofread by MDPI English Editing Service.
The reviewer wrote: “The experimental method to measure the temperature dependence of crack growth rate was not very well described. Also, to discuss the role of the temperature dependence of hydrogen-related properties on the crack growth rate, it is necessary to include the temperature dependence data of the crack growth without hydrogen. In that way, the readers would understand the effect of hydrogen and strong need for the modelling.".
Thanks for the comments. The description of experimental method had been added to the revised manuscript and read as “The experiments were carried out in C-2 or NOWATW solutions. The specimen was gritted to 9 mm thickness and a notch in the compact tension (CT) specimens was orientated in a direction perpendicular to the mode I loading direction. The dimensions of the specimen were 50× 50 × 9 mm. The specimen was polished to 600 grit finish and then pre-cracked in air by fatigue to generate a sharp crack from notch. Afterwards, the specimen was sealed in the test cells and pin-hole loaded with computer load control.”
The reviewer is absolutely right that temperature dependence of crack growth without hydrogen should also be introduced to prove the significance effect of hydrogen in the current study. Since the physical property and crack resistance of steel at different temperature have been well studied for many decades and some universal conclusions have been reached, a brief introduction has been added in the introduction section of the article and read as “When hydrogen is not present in steel, the toughness of the steel would increase with temperature, meanwhile, the hardness would decrease. Namely, the crack resistance of steel would increase with temperature[1,2]. However, when hydrogen is introduced, the temperature dependence of crack resistance of steel remains debatable and further studies are required in this field.”
The reviewer wrote: “Also, it is necessary to compare the data from all the different steels and to discuss the origin of difference in them based on the model developed. For example, the difference in crack growth rate under hydrogen absorption between different steels should be explained.".
Thanks for reviewer’s constructive comments. The difference of hydrogen absorption in different steels has been incorporated in the revised manuscript and read as “The size of plastic zone is smaller for a steel that processes higher yield strength. The plastic zone is the area concentrated with defects caused by plastic deformations, which would trap significant amount of hydrogen atoms. The theoretical model was based on HEDE theory; therefore, the trapping effects of plastic deformations caused defects had been ignored. Hence, for a high strength steel, in which the plastic zone is small, the model is more accurate and the fixing parameter n is approximating unity.”
The reviewer wrote: "The criteria of the hydrogen concentration at crack tip approximates one [15], as the free surface forms". Please rephrase the sentence so that the readers can understand the intention of the authors. Also please explain about the unit of 'one'."
Thanks for the comments. The manuscript has been revised as “As the atomic hydrogen concentration (an atomic ratio of number of hydrogen atoms to iron atoms) at the crack tip approximates 1, every bond between adjacent iron atoms would be weakened by hydrogen atom and the bindings are easily broken to form free surfaces fully covered by hydrogen atoms.”
The reviewer wrote: “NNpHSCC should be defined.”
Thanks for the comments. The NNpHSCC is defined as “Near neutral pH stress corrosion cracking” at the first place it appears in the revised manuscript.

Reviewer 3 Report
The English language is very poor and unacceptable for tthis scientific journal.
Author Response
The reviewer wrote: “The English language is very poor and unacceptable for tthis scientific journal."
Thanks for the suggestions. The revised manuscript has been polished and proofread by MDPI English Editing Service. In addition, several places in the manuscript has been rewritten and highlighted in the revised manuscript, which includes experimental method, the original difference in the models, the criteria of hydrogen concentration, etc.

Round 2
Reviewer 2 Report
The issues raised by the review appear properly addressed by the authors. Therefore, the publication of the manuscript is recommended.
Reviewer 3 Report
Very trivial and simple paper with plenty of language mistakes not suitable for a scientific journal. The scientific terms are mainly not correct and theory of fatigue and HE are very weak, e.g. cyclic plasticity is not considered at all.